# Effects of Resource Availability and Antibiotic Residues on Intestinal Antibiotic Resistance in *Bellamya aeruginosa*

**DOI:** 10.3390/microorganisms11030765

**Published:** 2023-03-16

**Authors:** Yayu Xiao, Peiyu Zhang, Huan Zhang, Huan Wang, Guo Min, Hongxia Wang, Yuyu Wang, Jun Xu

**Affiliations:** 1School of Ecology and Nature Conservation, Beijing Forestry University, Beijing 100083, China; 2Donghu Experimental Station of Lake Ecosystems, State Key Laboratory of Freshwater Ecology and Biotechnology of China, Institute of Hydrobiology, Chinese Academy of Sciences, Wuhan 430072, China; 3State Key Laboratory of Marine Resource Utilization in South China Sea, Hainan University, Haikou 570228, China

**Keywords:** florfenicol, intestinal microbiota, antibiotic resistance genes, pathogenic host, metabolism

## Abstract

Widespread and inappropriate use of antibiotics has been shown to increase the spread of antibiotics and antimicrobial resistance genes (ARGs) in aquatic environments and organisms. Antibiotic use for the treatment of human and animal diseases is increasing continuously globally. However, the effects of legal antibiotic concentrations on benthic consumers in freshwater environments remain unclear. In the present study, we tested the growth response of *Bellamya aeruginosa* to florfenicol (FF) for 84 days under high and low concentrations of sediment organic matter (carbon [C] and nitrogen [N]). We characterized FF and sediment organic matter impact on the bacterial community, ARGs, and metabolic pathways in the intestine using metagenomic sequencing and analysis. The high concentrations of organic matter in the sediment impacted the growth, intestinal bacterial community, intestinal ARGs, and microbiome metabolic pathways of *B. aeruginosa*. *B. aeruginosa* growth increased significantly following exposure to high organic matter content sediment. Proteobacteria, at the phylum level, and *Aeromonas* at the genus level, were enriched in the intestines. In particular, fragments of four opportunistic pathogens enriched in the intestine of high organic matter content sediment groups, *Aeromonas hydrophila*, *Aeromonas caviae*, *Aeromonas veronii*, and *Aeromonas salmonicida*, carried 14 ARGs. The metabolic pathways of the *B. aeruginosa* intestine microbiome were activated and showed a significant positive correlation with sediment organic matter concentrations. In addition, genetic information processing and metabolic functions may be inhibited by the combined exposure to sediment C, N, and FF. The findings of the present study suggest that antibiotic resistance dissemination from benthic animals to the upper trophic levels in freshwater lakes should be studied further.

## 1. Introduction

Antibiotics are used extensively in therapeutic medicines, disease prevention treatments, and as animal growth promoters [1,2,3], and their use continues to increase. Recent research has shown that global antibiotic consumption increased by 65% and antibiotic consumption in China increased by 79% between 2000 and 2015, and such a growth is projected to continue until 2030 [4]. It has been demonstrated that antibiotic residues in aquatic environments can pose ecological threats to aquatic organisms [5,6]. The presence of antibiotics can promote the emergence of antibiotic-resistant bacteria (ARB) and antibiotic resistance genes (ARGs) in aquatic environments [7,8]. In addition, the human-induced lake eutrophication increased microbial activity and antibiotic residue content, as well as the spread of ARB and ARGs [9,10], which increase the risk of bacterial drug resistance in aquatic organisms and in humans. According to predictions based on the current rates of antibiotic-resistance-associated deaths, there would be one such death every three seconds by the year 2050 [11]. Therefore, over the past few decades, the number of research articles addressing the topic has increased globally, with shifts in the intestinal microbiome being the current focus of research.

Pathogenic bacteria, such as *Microbacterium*, *Parachlamydiaceae*, and *Plesiomonas*, proliferate in the intestine of *Oreochromis niloticus* following feeding on oxytetracycline [12]. Previous studies have examined the effect of residual antibiotics in the aquatic environment on intestinal microbiology and ARGs in aquatic organisms [13]. The effects of common antibiotics, such as oxytetracycline, tetracycline, sulfamethoxazole, and FF, on the intestines of fish have been studied previously [12,13,14,15,16,17]. However, our understanding of how the intestinal microbes and ARGs of aquatic organisms respond to the combined effects of sediment and antibiotics remains limited. The intestine has the greatest microbial diversity in organisms. Intestinal microbes are essential for digestion, metabolism, and immune activities in all animals [18,19,20,21,22]. In addition, aquatic environments are continuously exposed to antibiotics owing to persistent residues in the environment [23,24]. The resulting development of drug-resistant microbes, which are the primary ARG hosts, and ABR bacteria development in aquatic environments, increase the risks of gastrointestinal illnesses caused by such bacteria [25]. Both symbiotic and pathogenic microbiota are affected by antibiotic exposure [26]. It has been reported that water input is a primary factor influencing microbial community structure in the intestine of *Poecilia reticulata* in the Uberabinha River, Brazil, with a positive correlation between ARGs and the dominant genera in intestinal samples, and most microbes are potential ARG hosts [26]. Low-dose florfenicol (FF) exposure causes dysbiosis in host microbiota [21]. With growing concerns over the presence of antibiotics and ARGs in aquatic environments globally, the use of most antibiotics in aquaculture is no longer permitted [14]. Currently, FF is one of the most commonly used antibiotics in aquaculture in most countries [27]. However, FF has been detected in aquaculture environments in recent years in China, for example, in Taihu Lake [28], coastal seawater in Dalian [29], and in the Guangzhou aquaculture area [30]. Therefore, FF was selected as a representative antibiotic in the present study. *Bellamya aeruginosa* is widespread in lakes and ponds in the middle and lower reaches of the Yangtze River. Nearly 30,000 tons of *B. aeruginosa* is harvested annually in Chaohu Lake, China, where it is mainly used as human food and for crab farming [31]. *B. aeruginosa* is a freshwater snail that is and play the role of primary consumer in aquatic food web [32]. However, to the best of our knowledge, the consequences of FF exposure, particularly on *B. aeruginosa* and its intestinal health and ARGs, has not been investigated comprehensively.

In the present study, *B. aeruginosa* was exposed to legal doses of FF for 12 weeks. The objective of the present study was to investigate the effect of FF and sediment with different nutrients on *B. aeruginosa* growth, as well as changes in intestinal health and ARG abundance in the host, using metagenomics. We report the impact of organic sediment matter, low-dose antibiotic residues, ARGs, and pathogenic hosts on the *B. aeruginosa* intestinal microbiome structure and function. The findings of the present study could enhance our understanding of food availability to aquatic organisms and antibiotics abundance in natural lakes, in addition to facilitating health risk assessments.

## 2. Materials and Methods

### 2.1. Antibiotics and Exposure

FF was purchased from Shandong Dexin Biology Science and Technology Co., Ltd., (Binzhou, China) and commercial feed was purchased from Cangzhou Zhengda Biological Products Co., Ltd. (Gangzhou, China). We collected approximately 800 *B. aeruginosa* individuals from Liangzi Lake, Hubei province, China. These organisms were acclimated in four 80-L tanks with dechlorinated water. *B. aeruginosa* in each tank received oxygen and were fed a commercial feed (Appendix A). We also collected sediments from Liangzi Lake to lay out in the experiment tanks. The initial properties of the sediment were 3.01 ± 0.79 mg/g organic phosphorus (P), 0.00 ± 0.01 mg/g organic nitrogen (N), and 16.60 ± 2.11 mg/g organic carbon (C). According to the methods in a previous study [33], submergent and emergent plants from Liangzi lake were oven dried (105 °C), ground, and then rewetted and used as organic matter to be added to the sediment. Two sediment treatments were set up, including one with high organic matter concentration (1.51 ± 0.32 mg/g organic N and 30.46 ± 2.83 mg/g organic C) and one with initial concentrations (0.00 ± 0.01 mg/g organic N, and 16.60 ± 2.11 mg/g organic C). There was no statistically significant change in organic P levels between the two sediment treatments following the aforementioned sediment treatment. Treated sediments (5-cm layer) were put in tanks (40 cm × 40 cm × 50 cm). The tanks were then placed in an open area, supplemented with 40 L of dechlorinated water, and allowed to settle for seven days.

*B. aeruginosa* with an initial mean weight of 1.18 ± 0.19 g were separated randomly into 20 tanks with 10 *B. aeruginosa* individuals per tank. The added FF was 10 mg/g body weight. The control and experimental groups were as follows: five tanks for FF added with high sediment organic C, organic N (HA), five control tanks with high sediment organic C, organic N (HN), five tanks for FF added with low sediment organic C, organic N (LA), and five control tanks with low sediment organic C, organic N (LN).

Over an 84-day study period, *B. aeruginosa* were fed commercial feed (once every fortnight). Considering that plants have the ability to absorb antibiotics, floating, leaf-floating, submerged plants, and attached algae on tank walls were removed daily throughout the study period. The pH, dissolved oxygen, total N (TN), and total P (TP) were maintained at 8.92 ± 0.57, 10.55 ± 1.76 mg/L, 1.57 ± 1.17 mg/L, and 0.01 ± 0.00 mg/L.

### 2.2. Sample Collection and Chemical Analysis

All surviving *B. aeruginosa* were collected at the end of the experiment and the number of survivors per tank and their final body weights were determined, which were used to calculate the survival rates and weight gain, respectively. The intestinal contents of *B. aeruginosa* were extracted and stored at −80 °C for DNA analysis. In addition, water samples were collected from each tank for antibiotic analysis.

FF concentration was determined using liquid chromatography–mass spectrometry (Waters Xevo TQ-S, Milford, MA, USA). The FF standards were purchased from Dr. Ehrenstorfer GmbH (Augsburg, Germany). Water samples (500 mL) were filtered through a 0.45-m membrane filter before being applied to an Oasis HLB cartridge (200 mg, 6 mL, Waters, Milford, MA, USA) for solid-phase extraction, as previously described [34]. The eluates were then exposed to a gentle N stream (Termovap Sample Concentrator, NK200-18, MIULAB, Hangzhou, China). A final volume of 1 mL was obtained by adding 10% acetonitrile. The samples were then analyzed using a Xevo TQ-S tandem quadrupole mass spectrometer (Waters, Milford, MA, USA).

Sediment organic carbon was pretreated with 1 mol/L HCL; then we used an elemental analyzer (Flash 2000, ThermoFisher Scientific, Waltham, MA, USA) to determine its value [35]. Sediment organic nitrogen represented the total nitrogen because organic nitrogen makes up ≥90% of the total N [36]. Organic P was measured by the content difference of the burned sample at high temperatures (550 °C) minus unburned sample detected in UV spectrophotometry [37].

### 2.3. DNA Extraction and Metagenomic Sequencing

The E.Z.N.A.^®^ Soil DNA Kit was used to extract metagenomic DNA, according to the manufacturer’s instructions (Omega Bio-Tek, Norcross, GA, USA). The DNA sample purity and concentration were evaluated using a NanoDrop2000 UV-Vis spectrophotometer (ThermoFisher Scientific) and a TBS-380 fluorometer (Turner Biosystems, Sunnyvale, CA, USA), respectively. DNA integrity was examined by electrophoresis on 1% agarose gel. Amplicon libraries were created after the DNA was broken up, using the Nexflex Rapid DNA-Seq Kit (Bioo Scientific, Austin, TX, USA). Amplicons were sequenced on an Illumina NovaSeq platform (Wuhan Baiaoweifan Biotechnology Co., Ltd., Wuhan, China).

Fastp (https://github.com/OpenGene/fastp, version 0.20.0) was used to remove reads of <50 bp, quality < Q20, and bases beginning with N. BWA (http://bio-bwa.sourceforge.net, version 0.7.9a, accessed on 18 November 2022) was used to remove reads from the host genome. MEGAHIT (https://github.com/voutcn/megahit, version 1.1.2) was used to assemble the optimized reads. Contigs with lengths ≥100 bp were selected for use in gene prediction and annotation. Taxonomic annotations of amino acid sequences from non-redundant gene sets were compared to the RefSeq non-redundant proteins database using Diamond (http://www.diamondsearch.org/index.php, version 0.8.35, accessed on 21 November 2022). ARG access data were obtained from the Comprehensive Antibiotic Resistance Gene Database (version 3.0.9).

### 2.4. Statistical Analysis and Network Analysis

The results of the experiment were expressed as the mean ± standard error of the mean. A *t*-test was used to determine whether there were any significant differences between treatments. R (version 4.0.3) was used to visualize the data. Diversity was assessed and displayed using the R vegan package. The R software (v. 4.0.3) was used to create bar plots, heatmaps, circular bar plots, circular plots, and to perform the correlation analysis. Reads per kilobase per million mapped reads (RPKM) were used to calculate the relative abundances of bacteria and ARGs. Statistical significance was determined at *p* ≤ 0.05 for the aforementioned analyses.

## 3. Results

### 3.1. Effect of Dietary Florfenicol and Sediments on B. aeruginosa Growth

The growth of *B. aeruginosa* was not significantly affected by dietary FF after 12 weeks of antibiotic treatment. Exposure to FF (HA and LA) did not significantly slow growth performance when compared with the controls (HN and LN) (Table 1). However, *B. aeruginosa* in the two sediment groups differed significantly in terms of weight gain (Table 1). *B. aeruginosa* grew slowly in sediment with low organic matter.

### 3.2. B. aeruginosa Intestinal Microbiome Structure after Treatments

Proteobacteria, Firmicutes, Tenericutes, Bacteroidetes, and Actinobacteria were the top five most abundant phyla in the annotation results of all exposure groups following treatment (Figure 1A). To further explore the changes in the microbial structure, the relative abundances at the genus level were determined (Figure 1B). The most abundant species at the genus level included *Aeromonas*, *Bacillus*, *Clostridium*, *Dechloromonas*, *Mycoplasma*, *Polynucleobacter*, *Proteocatella*, *Pseudomonas*, *Tolumonas*, and *Vibrio*. All treatment groups could be divided into two groups, with the microbial communities of the HN and HA exposure groups placed in one category, with more abundant species. LN and LA exposure groups were placed in other category, with less abundant species. The microbial communities differed at the phylum and genus levels between the high organic matter content sediment (HN and HA) and the low organic matter content sediment (HN and HA) exposure groups, but not between the no-FF (HN and LN) and FF (HA and LA) exposure groups. *Aeromonas*, in particular, was detected in all groups and was greatly enriched in the high organic matter content sediment exposure group (HN and HA). It was categorized as a Proteobacterium at the phylum level, whereas at the species level, the genus *Aeronomas* contains numerous pathogens.

### 3.3. B. aeruginosa Intestinal Antibiotic Resistome Structure after Treatments

A total of 286 antibiotic ARG subtypes were detected in the intestines of *B. aeruginosa* and classified into 23 types. Organic matter in the sediments altered the ARG distribution in the intestine significantly (Wilcoxon test, *p* < 0.05), with an increase in the number of ARGs (Figure 2A). Figure 1B shows the 21 types of ARGs, namely aminoglycoside, bacitracin, beta-lactam, bleomycin, carbomycin, chloramphenicol, fosfomycin, fosmidomycin, kasugamycin, macrolide-lincosamide-streptogramin (MLS), multidrug, polymyxin, puromycin, quinolone, rifamycin, sulfonamide, tetracenomycin_C, tetracycline, trimethoprim, vancomycin, and unclassified ARGs. The top ARG types of all samples were multidrug and MLS ARGs. When compared with that in the LN and HN samples, the relative abundance of ARG types did not change significantly following FF exposure (LA and HA). In addition, the relative abundances of ARG types did not differ significantly between the high and low nutrient organic matter level in the sediment treatments. ARG abundance was enriched in HA and HN at the ARG subtype level following nutrient treatments (Figure 2C). Among the top 50 detectable ARGs, MLS and multidrug ARGs were the most frequently detected types. There were 10 ARGs and 13 ARGs classified as MLS and multidrug ARGs, respectively.

### 3.4. Pathogenic Hosts of ARGs in B. aeruginosa Intestines

ARG-carrying genes (1618) were identified in the intestinal bacteria of *B. aeruginosa*. ARGs encoding resistance to aminoglycoside, bacitracin, beta-lactam, carbomycin, chloramphenicol, fosmidomycin, kasugamycin, MLS, multidrug, polymyxin, sulfonamide, tetracenomycin_C, tetracycline, trimethoprim, and vancomycin were among the top 50 genes. ARG-carrying genes were annotated as fragments of Proteobacteria in the HN, HA, and LN groups (54%, 76%, and 100%, respectively). Additionally, 92% of the ARG-carrying genes in the LA group were annotated as Firmicute fragments (Figure 3). Using the pathogen–host interactions database and a previously summarized pathogen list [38], 13 ARG-carrying genes were identified as pathogen-host-carried genes (Table 2). The pathogen fragments carrying ARGs included *aac6-I*, *aadE*, *chloramphenicol_exporter*, *cat_chloramphenicolacetyltransferase*, *macB*, *vatB*, *vatE*, *tcmA*, *bcrA*, and *mexT*. Two pathogen fragments carried an ARG that encoded resistance to chloramphenicol, two pathogen fragments carried an ARG that encoded resistance to aminoglycoside, and six pathogen fragments carried an ARG that encoded resistance to MLS. The remaining three ARG-carrying pathogen pieces encoded resistance to bacitracin, multidrug, and tetracenomycin_C, respectively. Notably, *Aeromonas hydrophila*, which was grouped in the HN and HA groups, was often found in the *B. aeruginosa* intestine. *A. hydrophila* is a notoriously difficult-to-treat pathogen that can cause severe disease and infection in the intestines of aquatic organisms.

### 3.5. High Organic Matter Content Sediments Altered the Intestinal Microbe Function

Different intestinal bacterial metabolic pathways produce different metabolites that influence all aspects of host physiological functions. The functional pathways of the microbial communities were inferred using the Kyoto Encyclopedia of Genes and Genomes (KEGG) database to understand the functional profiles of the intestinal bacterial community after treatment. In total, 8322 functional pathways were identified in the intestine of *S. aeruginosa*, (Figure 4). The four groups shared the majority of the functional pathways. Upon calculating the fold difference between the high (HA and HN) and low organic matter content groups (HN and LN), 27 functions were observed to be upregulated (Figure 4A). When the fold difference between no FF exposure (LN and HN) and FF exposure (LA and HA) was calculated, all functional pathways showed no significant differences (Appendix A). Sediment organic matter levels caused functional changes in the intestinal flora rather than FF. Twenty-seven altered functional pathways from 14 categories were selected for the analysis of two treatment factors (Figure 4B). Folding, sorting and degradation function, glycan biosynthesis and metabolism function, and protein families: genetic information processing function were significantly positively correlated with sediment organic matter, whereas the replication and repair function was significantly negatively correlated with the sediment organic matter level. Two unclassified functions (genetic information processing and metabolism) were significantly negatively correlated with FF exposure.

## 4. Discussion

Food sources and environmental conditions can influence the growth of benthic consumers. The vast majority of benthic consumers consume surface sediment [39]. C and N from food can theoretically be stored at homeostatic states in body tissues [40]. Sedimentary organic matter fuels the benthic food chain and is an important recycler in lake ecosystem energy flows, as well as the C and N cycles [41,42]. In the present study, exposure to low-nutrient sediments suppressed *B. aeruginosa* weight gain. Conversely, exposure to FF in the diet at legal doses did not result in such a phenomenon, which is consistent with the findings of some studies on aquatic animals, such as *Gadus morhua* [43], *Oreochromis* sp. [44,45], and *Oreochromis niloticus* [46]. In addition, FF can be degraded in an open area by water temperature (10 °C), electrolytes, and UV processes [47,48,49]. This implies that *B. aeruginosa* growth may be affected more by sediment organic matter (C and N) than by the low doses of antibiotics applied in the present study.

*B. aeruginosa* inhabits the sediment–water interface [50]. It uses organic matter (particularly C and N) in sediments to facilitate energy exchange and growth [40,51]. The composition of the diet shapes the composition of intestinal microbiota and can integrate new genes into the microbiota of the intestine [52]. For example, *Eriocheir sinensis* intestinal microbiota were composed of bacteria harbored by its food source [53]. Based on these findings, diet modulation could be a potential treatment for dysbiosis caused by antibiotic exposure, such as supplying bee pollen in food to develop the intestinal tract of African catfish [54]. In the present study, organic matter sediment influenced intestinal microbiota community abundance. Microbial community alteration is the major driver of ARG distribution [55]. The present findings showed a potential correlation between Proteobacteria abundance and increased ARG abundance, which is consistent with previous research that discovered Proteobacteria as potential ARG hosts [56]. An increase in Proteobacteria abundance, moreover, is an indicator of dysbiosis. Furthermore, changes in intestinal microbiota may influence various physiological processes [57]. ARGs (*aac6-I, aadE, chloramphenicol_exporter, cat_chloramphenicol_acetyltransferase, macB, vatB,* and *tcmA*) were found in the Proteobacteria fragments in the present study. *A. hydrophila*, *Aeromonas caviae*, *Aeromonas veronii*, and *Aeromonas salmonicida* were detected as ARG-carrying pathogens and are fairly resilient pathogens. They can produce enterotoxins, posing a significant challenge to host microbiota stability and resilience [58,59]. Such hosts can spread ARGs to confer resistance to antibiotics via horizontal gene transfer and mutational events [60,61], resulting in the spread of ARB and increasing the economic burdens in aquaculture [62].

Increased organic matter concentration in sediments is undesirable. In our study, high organic matter sediment enhanced Proteobacteria abundance at the phylum-level and *Aeromonas* abundance at the genus level. The frequency of pathogen detection in the intestine increases dramatically following exposure to high-nutrient sediments. ARGs in the gut are under selective pressure from organic matter from food, in addition to microbial interactions and effects [63]. It has been demonstrated that sediment organic C influences resistance gene distribution in benthic animals [64]. In another study, the intestinal microbiota in two species of wild crabs served functions comparable to those of sediment [65]. The two ARGs most frequently observed in our samples were MLS and multidrug ARGs. Eutrophic lakes have also been shown to contain a range of ARGs, primarily multidrug ARGs [66]. ARGs can co-occur, and co-selection is likely to enrich the resistance of ARB to unrelated antibiotics [67,68]. Therefore, antibiotic resistance in eutrophic water environments may pose a significant challenge for manipulating the microbiota against ARB. Additionally, our results demonstrate that the metabolic activity of intestinal bacteria increases with an increase in organic matter contents. Three functions actively related to protein processing and lipid metabolism demonstrated a strong correlation with C and N, suggesting that *B. aeruginosa* has undergone certain dietary adaptations [69,70,71,72,73,74]. Similarly, oyster gut microbiomes respond to higher nutrient levels in the diet by upregulating certain glucose and lipid metabolism activities [75]. However, because sediment C and N have the capacity to suppress DNA transcription, the rise in sediment C and N in the present study has the potential to result in DNA damage. In our study, the metabolic expression of resistance was unaffected by short-term exposure to low-dose FF. Following exposure to our experimental conditions, the metabolic function pathways of the gut microbiota did not significantly change [76]. Furthermore, two unidentified processes associated with the processing of genetic information and metabolism were inhibited following exposure to FF under increasing sediment C and N. Further research should be conducted on the joint effects of eutrophication and antibiotic exposure on microbial structures and their functions.

## 5. Conclusions

In conclusion, the organic matter in the sediment facilitated *B. aeruginosa* proliferation. High organic matter sediments affected the intestinal microbiota and metabolic expression of *B. aeruginosa*, according to metagenomic sequencing analysis results. Some processes were activated to adjust to the dietary organic materials. However, exposure to sediment C, N, and FF may disrupt several processes involved in metabolism and processing of genetic information. Additionally, exposure to high levels of organic material dramatically enhanced the presence of pathogens harboring ARG, which was linked to increases in the abundance of the genus *Aeromonas*. The findings of the present study enhance our understanding of the risks that antibiotics and eutrophic sediments pose to aquatic life. Furthermore, additional experimental validation can be performed to determine the impact of the combined exposure to sediments C, N, and FF on benthic animal intestinal functioning.

## Figures and Tables

**Figure 1 microorganisms-11-00765-f001:**
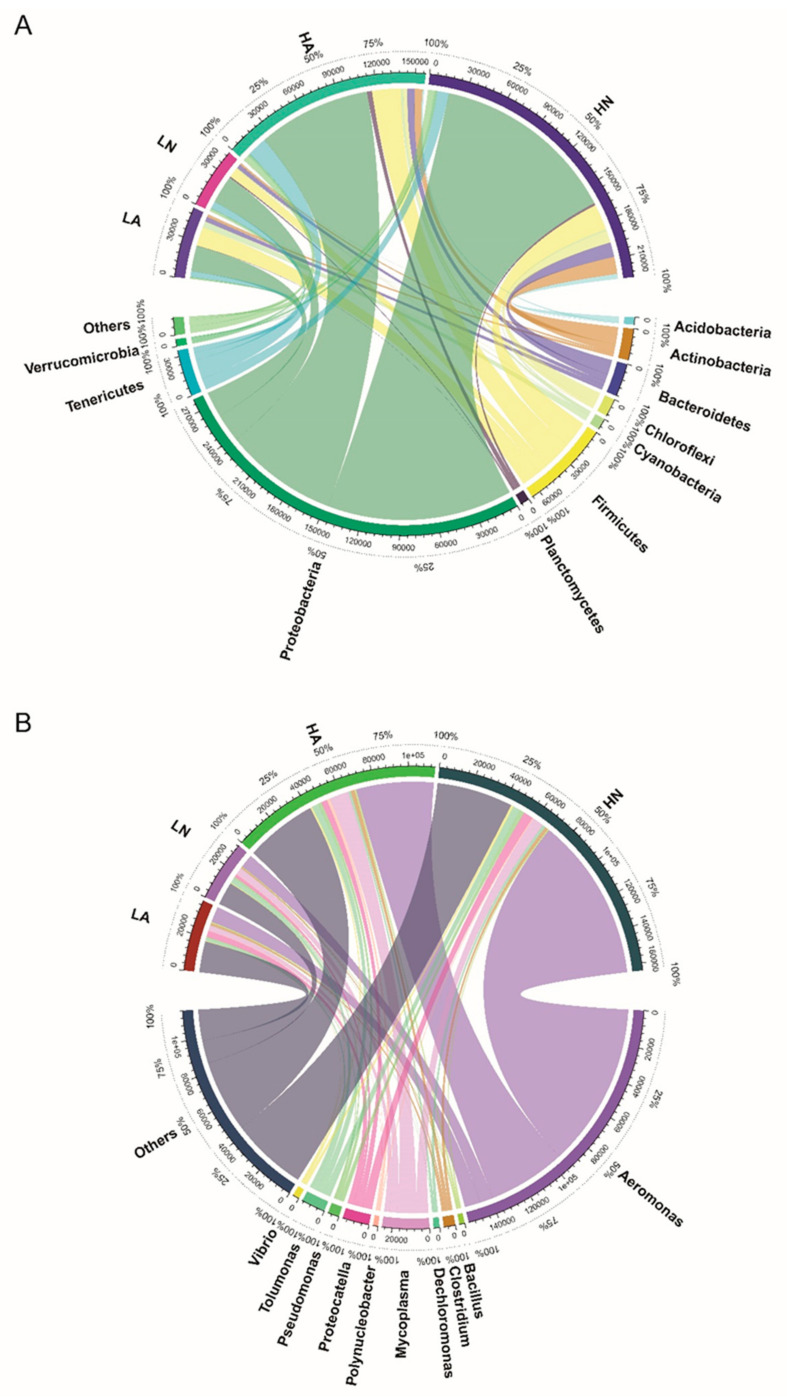
Intestinal microbiota composition of *Bellamya aeruginosa* at the phylum level (**A**) and genus level (**B**).

**Figure 2 microorganisms-11-00765-f002:**
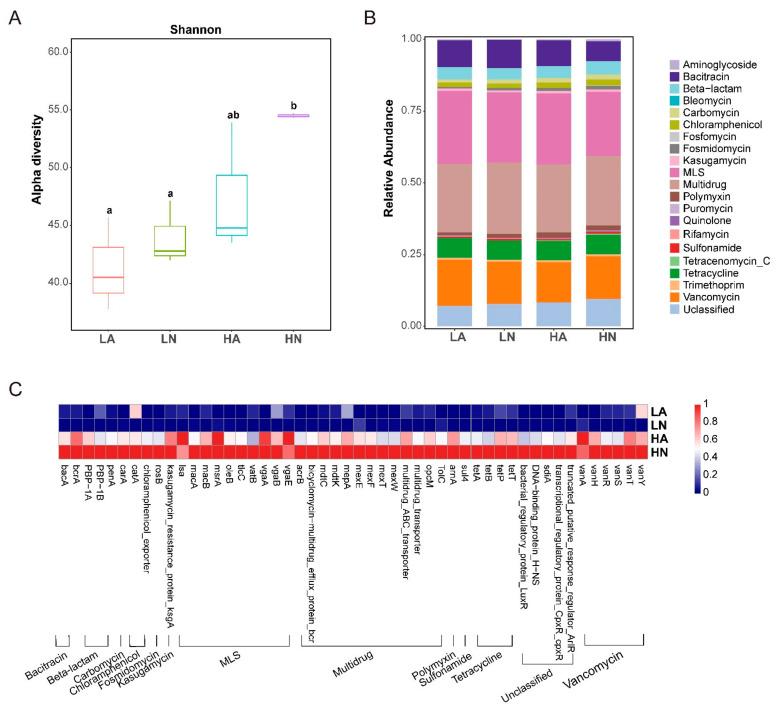
Diversity and abundance of different antibiotic resistance genes (ARGs). (**A**) Shannon index of ARG alpha diversity (Wilcoxon test, *p* < 0.05, the significance of the difference was indicated by letters). (**B**) The relative abundances of different ARG types. (**C**) The profile of the top 50 ARGs obtained by metagenomic sequencing.

**Figure 3 microorganisms-11-00765-f003:**
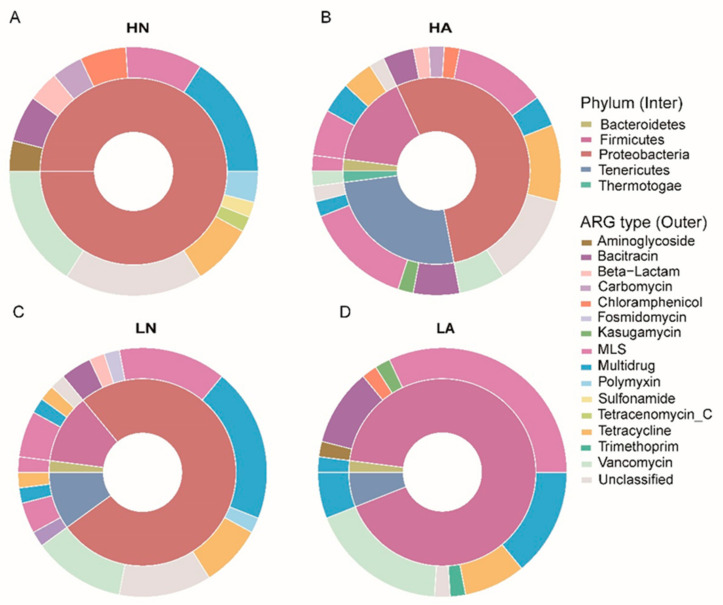
Annotation of antibiotic resistance gene-carrying hosts at the phylum level. Here, only the 50 most abundant genes are shown. (**A**) HN group (**B**) HA group (**C**) LN group (**D**) LA group.

**Figure 4 microorganisms-11-00765-f004:**
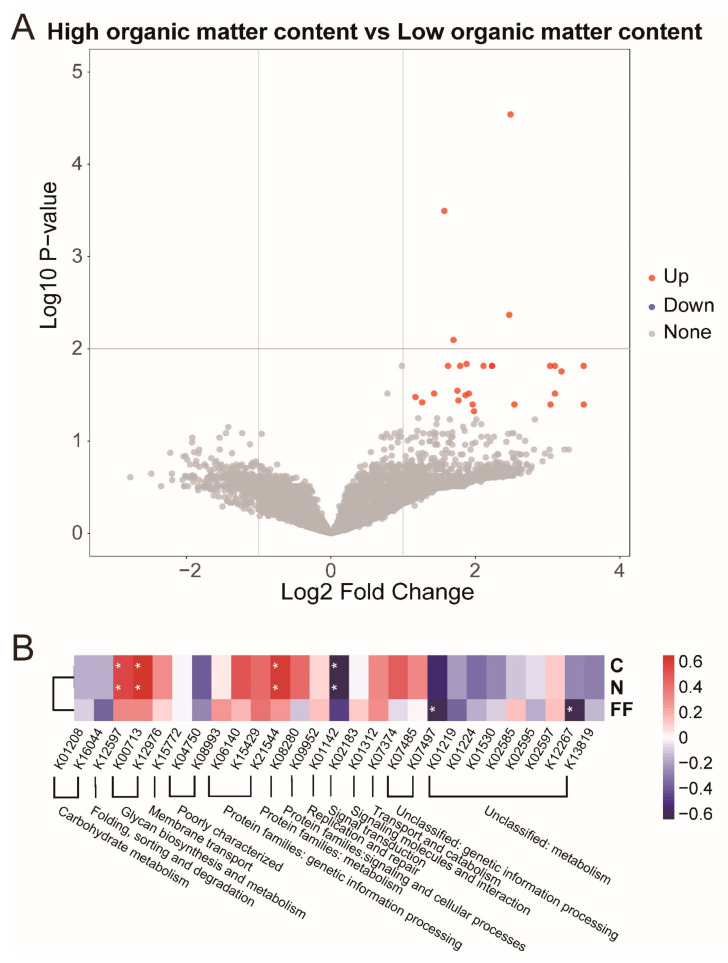
The effect of florfenicol and sediment nutrients on functional pathways of intestinal bacteria were analyzed by using the Kyoto Encyclopedia of Genes and Genomes database. (**A**) Impacts of sediment organic matter levels on the functional pathways. (**B**) Heatmap of the upregulated functional pathways. * Represents statistically significant differences at *p* < 0.05.

**Table 1 microorganisms-11-00765-t001:** Effects of two sediment types and florfenicol diets on *Bellamya aeruginosa* growth.

	HN	HA	LN	LA
Initial weight (g)	1.222 ± 0.186	1.199 ± 0.194	1.143 ± 0.197	1.163 ± 0.157
Final weight (g)	1.826 ± 0.265 ^a^	1.843 ± 0.261 ^a^	1.488 ± 0.242 ^b^	1.527 ± 1.0.305 ^b^
Weight gain (%)	49.745 ± 15.663 ^a^	53.569 ± 8.395 ^a^	30.363 ± 16.9 ^b^	31.451 ± 17.67 ^b^
Survival rate (%)	97.000 ± 0.0270 ^a^	99.000 ± 0.0220 ^a^	97.500 ± 0.035 ^a^	95.500 ± 0.037 ^a^

Data are expressed as the mean ± standard error of the mean (n = 10). A significant difference (*p* < 0.05) is indicated by the superscript in the same row.

**Table 2 microorganisms-11-00765-t002:** Annotation of the top 50 antibiotic resistance gene-carrying genes at the species level.

Gene_Id	Species	ARGs Carried	Treatment
3.300787_5	*Aeromonas hydrophila*	aminoglycoside~*aac6-I*	HN
3.233839_2	*Aeromonas hydrophila*	aminoglycoside~*aadE*	HN
3.88430_7	*Aeromonas hydrophila*	chloramphenicol~*chloramphenicol_exporter*	HN
2.254851_21	*Aeromonas hydrophila*	chloramphenicol~*cat_chloramphenicol_acetyltransferase*	HN
2.49360_53	*Aeromonas hydrophila*	MLS~*macB*	HN
2.57139_23	*Aeromonas hydrophila*	MLS~*macB*	HN
3.55596_20	*Aeromonas hydrophila*	MLS~*vatB*	HN
2.237501_4	*Aeromonas hydrophila*	tetracenomycin_C~*tcmA*	HN
3.60413_2	*Aeromonas caviae*	bacitracin~*bcrA*	HA
3.55596_20	*Aeromonas hydrophila*	MLS~*vatB*	HA
4.467441_4	*Aeromonas veronii*	MLS~*vatE*	HA
2.49360_53	*Aeromonas hydrophila*	MLS~*macB*	HA
3.59927_2	*Aeromonas salmonicida*	multidrug~*mexT*	LN

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
