# Peer review of "Effects of Resource Availability and Antibiotic Residues on Intestinal Antibiotic Resistance in Bellamya aeruginosa"

_microorganisms, 2023, doi:10.3390/microorganisms11030765_

Round 1
Reviewer 1 Report
General overview:-
This manuscript provides interesting data to enhance and understanding of food availability to aquatic organisms and antibiotics abundance in natural lakes, in addition to facilitating health risk assessments.; this work is really much needed because it doesn’t cause chemical pollution of environment which is human harmful.
Grammatical and language content:
I think that this manuscript is good in its overall quality. However, further revision is needed.
Scientific content:
Authors must improve abstract
The method description is not clear and accurate enough
Results as well as discussion are good.
Author Response
Response: We appreciate your professional review work on our paper. We have modified abstract and method according to your comment and the revised parts had been marked with red in the submitted manuscript.

Reviewer 2 Report
How antibiotic in aquatic environment will impact organisms if of great concern worldwide. This study analyzed the influence of florfenicol exposure and sediment nutrient in Bellamya aeruginosa intestinal bacterial community and ARGs. The topic is suit for publish in Microorganisms. But in abstract, introduction and method parts need some improvements as following:
1. In Abstract, “sediment organic matter (carbon [C] and nitrogen [N])” should be “sediment carbon [C] and nitrogen [N]”. In Method 2.2 you mentioned “C and N in the sediment were measured using an elemental analyzer (Flash 2000, ThermoFisher Scientific, USA)”, this should be total C and total N in sediment. Revise to “C and N in sediment” in Line 19, 20-21, 24, as organic matter content in sediment” not exact.
2. In Introduction, delete ‘s’ after ‘ARGs’ in Line 44. Move third paragraph (Line 61-68) before second paragraph (Line50-60) would be clearer and more coherent.
3. Not clear about this “B. aeruginosa, is a freshwater snail at an intermediate trophic level in the aquatic food web and plays an important role in bioaccumulation”. Do you mean this snail is the primary consumer and play important role in aquatic food web?
4. In Method, revise Line 110- 114 as “The added FF was 10 mg/g body weight. The control and experimental groups were as follows: five tanks for FF added with high sediment C, N(HA), five control tanks with high sediment C, N (HN), five tanks for FF added with low sediment C,N (LA), and five control tanks with low sediment C, N(LN). ”.
5. Line 135 provide detailed method for this “P in the sediment was measured using standard meth-135 ods.”.
6. In Result Line 167-170 add cite for Table 1 as “Exposure to FF (HA and LA) did not significantly influence final weight when compared with the controls (HN and LN) (Table 1). However, B. aeruginosa in the two sediment groups differed significantly in terms of final weight and weight gain (Table 1).”.
Author Response
Response: Thanks for your comments for our manuscript. These comments are all helpful for improving our paper. We have made correction accordingly and marked the revision in red in revised manuscript. The main corrections in the paper are as following:
- In Abstract, “sediment organic matter (carbon [C] and nitrogen [N])” should be “sediment carbon [C] and nitrogen [N]”. In Method 2.2 you mentioned “C and N in the sediment were measured using an elemental analyzer (Flash 2000, ThermoFisher Scientific, USA)”, this should be total C and total N in sediment. Revise to “C and N in sediment” in Line 19, 20-21, 24, as organic matter content in sediment” not exact.
Response: Thanks for your suggestion. Before using an elemental analyzer to determine the amount of soil organic carbon, the samples were pretreated with 1 mol/L HCL for the purpose of measuring the sediment's organic matter (Min et al., 2020). Additionally, inorganic N makes up a relatively small percentage of the total N in the sediment samples, so we the N in the study mainly were organic N (Kpomblekou, 2006). Thus, we used “sediment organic matter (carbon [C] and nitrogen [N])” in this paper.
Reference:
Kpomblekou‐A K. Relative proportion of inorganic and total nitrogen in broiler litter as determined by various methods[J]. Journal of the Science of Food and Agriculture, 2006, 86(14): 2354-2362.
Min X, Wu J, Gao C, Xu L, Li L, Wang F. The Comparative Study on Different Pretreatment Methods of Soil Organic Carbon Determined by Elemental Analyzer [J]. Journal of Salt Lake Research, 2020, 28 (04): 64-70
- In Introduction, delete ‘s’ after ‘ARGs’ in Line 44. Move third paragraph (Line 61-68) before second paragraph (Line50-60) would be clearer and more coherent.
Response: Thanks for your careful checks. Revised accordingly.
- Not clear about this “B. aeruginosa, is a freshwater snail at an intermediate trophic level in the aquatic food web and plays an important role in bioaccumulation”. Do you mean this snail is the primary consumer and play important role in aquatic food web?
Response: We apologize for the phrase confused you. Bellamya aeruginosa is primary consumer in the aquatic food web, which could link primary producers and other consumers. We have revised the statement in Line 81-83.
- In Method, revise Line 110- 114 as “The added FF was 10 mg/g body weight. The control and experimental groups were as follows: five tanks for FF added with high sediment C, N(HA), five control tanks with high sediment C, N (HN), five tanks for FF added with low sediment C,N (LA), and five control tanks with low sediment C, N(LN). ”.
Response: Thank you for your constructive comments. We have carefully considered the suggestion and make change. The red part that has been revised as follows: The added FF was 10 mg/g body weight. The control and experimental groups were as follows: five tanks for FF added with high sediment organic C, organic N (HA), five control tanks with high sediment organic C, organic N (HN), five tanks for FF added with low sediment organic C, organic N (LA), and five control tanks with low sediment organic C, organic N(LN).
- Line 135 provide detailed method for this “P in the sediment was measured using standard meth-135 ods.”.
Response: Thank you for your reminder. We have revised the sentence as “Organic P was the content difference of P in sediment between by burned at high temperatures (550°C) and not burned detected in UV spectrophotometry [37].”
Reference:
Zhou J, Shen R. Dictionary of Soil Science [M]. Science Press, 2013.
- In Result Line 167-170 add cite for Table 1 as “Exposure to FF (HA and LA) did not significantly influence final weight when compared with the controls (HN and LN) (Table 1). However, B. aeruginosa in the two sediment groups differed significantly in terms of final weight and weight gain (Table 1).”.
Response: Thanks for your suggestion. We made corresponding corrections in Line 177-178.

Reviewer 3 Report
In my opinion, a very high quality MS, which should be accepted after minor revision. My few remarks are included in the text. To see them all, open the file in Acrobat Reader. In the Discussion section, I suggest expanding it with one thread related to the influence of food on the change in the intestinal microbiome. In general, MS reads very well, it is written clearly and transparently. It was a great pleasure to review this MS.

Author Response
Thank you for your comments concerning our manuscript entitled “Effects of resource availability and antibiotic residues on intestinal antibiotic resistance in Bellamya aeruginosa”. Those comments are all valuable and very helpful for revising and improving our paper. According to your suggestions, we made corresponding corrections and are given in the red text in new manuscript. The main detailed corrections are listed below.
For the comment 8 in pdf file: which one was the control group?
Response: A two-factor controlled experiment was designed for this study. There were four distinct groups (FF with sediment with high organic matter; sediment with high organic matter; FF with sediment with low organic matter; sediment with low organic matter). Thus, the two groups lacking FF can be considered control groups.
For the comment 13 in pdf file: add English and full Latin names
Response: To our knowledge, Bellamya aeruginosa has no specific English name. Most studies refer to it as a freshwater snail, which we believe does not adequately represent the organism. So, we used its Latin names in this study.
For the comment 15 in pdf file: In the Discussion section, You suggested expanded it with one thread related to the influence of food on the change in the intestinal microbiome.
Response: Thanks for your suggestion. We cited the paper you mentioned in the pdf file in our manuscript. We added in Line 294-296 to expand our discussion.
The corrections for the rest 13 comments are revised in Line 61, 63, 72, 97, 98, 132, 141, 147, 177, 181, 204, 299.
Thanks for your careful checks, we are sorry for our careless.
